# The Surface Water Chemistry (SWatCh) database: A standardized global database of water chemistry to facilitate large-sample hydrological research

Lobke Rotteveel[1], Franz Heubach[2], Shannon M. Sterling[1]

[1] Sterling Hydrology Research Group, Dalhousie University, Halifax, B3H 4R2, Canada
[2] Department of Mechanical Engineering, Dalhousie University, Halifax, B3H 4R2, Canada

*Correspondence to*: Shannon M. Sterling (shannon.sterling@dal.ca)

**Abstract.** Openly accessible global scale surface water chemistry datasets are urgently needed to detect widespread trends and problems, to help identify their possible solutions, and determine critical spatial data gaps where more monitoring is required. Existing datasets are limited in availability, sample
size/sampling frequency, and geographic scope. These limitations inhibit the answering of emerging transboundary water chemistry questions, for example, the detection and understanding of delayed recovery from freshwater acidification. Here, we begin to address these limitations by compiling the global surface water chemistry (SWatCh) database, available on Zenodo (DOI: 10.5281/zenodo.6484939). We collect, clean, standardize, and aggregate open access data provided by six national
and international programs and research groups (United Nations Environment Programme, Hartmann et al. (2019), Environment and Climate Change Canada, the United States of America National Water Quality Monitoring Council, European Environment Agency, and the United States National Science Foundation McMurdo Dry Valleys Long-Term Ecological Research Network), to compile a database containing information on sites, methods, and samples, and a GIS shapefile of site locations. We re-
move poor quality data (for example, values flagged as "suspect" or "rejected"), standardize variable naming conventions and units, and perform other data cleaning steps required for statistical analysis. The database contains water chemistry data for streams, rivers, canals, ponds, lakes, and reservoirs across seven continents, 24 variables, 33,722 sites, and over 5 million samples collected between 1960
and 2022. Similar to prior research, we identify critical spatial data gaps on the African and Asian continents, highlighting the need for more data collection and sharing initiatives in these areas, especially considering freshwater ecosystems in these environs are predicted to be among the most heavily impacted by climate change. We identify the main challenges associated with compiling global databases – limited data availability, dissimilar sample collection and analysis methodology, and reporting ambi-
guity – and provide recommended solutions. By addressing these challenges and consolidating data from various sources into one standardized, openly available, high quality, and trans-boundary database, SWatCh allows users to conduct powerful and robust statistical analyses of global surface water chemistry.

# 1 Introduction

Globally, 159 million people are reliant on untreated surface water, with only one in three people having access to safely managed drinking water services (World Health Organization and United Nations Children's Fund (WHO and UNCF), 2017). With two-thirds of the global population (4.0 billion people) already experiencing water shortages at least one month per year (Mekonnen and Hoekstra, 2016), and 4.8-5.7 billion people projected to experience water shortages by 2050 (Burek et al., 2016), maintaining the quality of drinking water sources is paramount to human health and society. One of the main obstacles to achieving this goal is a lack of openly available, high quality, transboundary data (WHO and UNCF, 2017). Existing large-sample water quality datasets have: 1) limited availability, for example, raw data may not be published with journal articles (Alsheikh-Ali et al., 2011); 2) limited sample size, for example, datasets may only include one water body type (Hartmann et al., 2014); or 3) limited geographic scope, for example, national datasets only include data for one country.

Delayed acidification recovery is an example of a transboundary problem which would benefit from a large-sample dataset. Here, we define a "transboundary problem" to be a water quality issue, or cause of a water quality issue, which crosses international borders. For example, a main driver of freshwater acidification in Atlantic Canada is acid deposition originating from all the major production regions in North America, including those in the United States of America (Shaw, 1979). A similar definition of "transboundary problem" is often used when discussing water availability issues which cross international borders (for example, Thu & Wehn, 2016). Ecosystem acidification and associated elevated aluminum (Al) concentrations are responsible for the loss of economically significant fish species (Committee on the Status of Endangered Wildlife in Canada, 2011; Dennis and Clair, 2012), reductions in crop success (Collignon et al., 2012), reduced forest health (Collignon et al., 2012; DeHayes et al., 1999; de Wit et al., 2010) and therefore carbon sequestration, and increased cost of water treatment (Letterman and Driscoll, 1988); further, high Al in drinking water resources may contribute to human osteological and neurological diseases (WHO, 2010). Prior large-sample (Björnerås et al., 2017; Monteith et al., 2007), and global scale (Weyhenmeyer et al., 2019) studies on freshwater acidification indicate that recovery is delayed in some regions. But, so far, there is no openly available global scale database of acidification related water chemistry which includes Al, increased concentrations of which are one of the most biotically toxic effects of acidification (Gensemer and Playle, 1999).

There is a need for harmonized large-sample hydrological research (Blöschl et al., 2019), and global datasets are required to develop global water chemistry models (Harrison, Caraco, et al., 2005; Harrison, Seitzinger, et al., 2005). The majority of water quality research has focused on catchment scale datasets, which narrows our understanding of hydrochemical processes to catchments which have historically been studied. Catchment scale analyses make valuable contributions to our understanding of hydrochemical processes. However, variability in catchment response to perturbation, which is potentially indicative of variability in hydrochemical processes, is difficult to evaluate in a robust manner without an approach which assesses multiple catchments/regions in a harmonized way. In the case of freshwater acidification, water chemistry response to acid deposition may be altered by geology and land use/land cover; thus, observations made in one watershed/region may not generalize to others (for example, Clair

et al., 2011; Rotteveel & Sterling, 2020). For example, watershed response to acid deposition is influenced by weak acids in regions with slow-weathering, base cation ($C_B$) poor, bedrock, but not in regions with higher $C_B$ geology (Clair et al., 2011; Stoddard et al., 1999), and watersheds with high-intensity forest harvesting may be more strongly affected by acid deposition than those with less disturbance (Aherne et al., 2008; Feller, 2005).

Obtaining and consolidating water chemistry datasets for transboundary hydrological research is challenging due to limited data access, and disparate (that is, dissimilar) data collection programs and data reporting formats. Access may be limited because data are not published and/or kept confidential, as is the case for some sites within the United Nations International Centre for Water Resources and Global Change's Global Water Quality Database and Information System (GEMStat). Data collection programs are dissimilar largely due to a lack of international variable and analysis method definitions (WHO and UNCF, 2017). For example, Al measurements may not be comparable across different functional, operational, and classical species definitions (Namieśnik and Rabajczyk, 2010; Ščančar and Milačič, 2006). Lastly, disparate variable naming conventions, units, and censored data notation complicates consolidation of datasets from different sources, as these notations must first be standardized.

Here, we aim to address the above limitations by contributing an openly available, standardized, easy-to-use, global water chemistry database. We focus on providing data to address the problem of delayed freshwater acidification recovery by collecting, cleaning, standardizing, and compiling datasets of acidification related water chemistry variables. Specifically, our research goals are 1) to develop a global database of acidification related surface water chemistry, 2) to identify the main limitations associated with compiling this database, 3) to identify and characterize critical spatial data gaps within existing datasets, and 4) to provide recommendations for data reporting and storage to facilitate its easy access and use by other researchers.

## 2 Methods

### 2.1 Data Sources

We obtained input data for SWatCh from openly available datasets published by national and international agencies and from datasets available on open-access servers (Table 1). Our search terms were "water chemistry data" or "water quality data" and "global" or a country name, as listed in the United Nations member countries (United Nations, 2009). Our data search did not have a geographic focus, although our sources were limited to datasets available in English. Datasets likely missed by this approach include those hosted on servers or websites without (or without English) Search Engine Optimization (SEO); that is, those which have not been optimized with keywords identifiable by search engines to provide results (Google, 2020). All datasets were originally downloaded in September 2019 and updated during the manuscript review process in April and March 2022 so newly published data could be included. The GloRiCh dataset was not re-downloaded because it not been updated since the prior download, and the National Water Quality Monitoring Council Water Quality Portal dataset was not re-downloaded due to unresolved internal server errors.

## 2.2 Data Inclusion

SWatCh includes 24 water chemistry variables collected in untreated surface water bodies. We define "untreated" as water that is not wastewater or receiving treatment plant effluent near to the sample collection site (for example, sites described as "wastewater" or "effluent"). The included water body types are streams, rivers, canals, ponds, lakes, and reservoirs. The included water chemistry variables are metals: Al, and iron (Fe); $C_B$'s: calcium (Ca), magnesium (Mg), potassium (K), and sodium (Na); other measures of buffering capacity: acid neutralization capacity (ANC), alkalinity, carbonate ($CO_3$), and bicarbonate ($HCO_3$); acid anions: sulfate ($SO_4$), nitrate ($NO_3$), and nitrite ($NO_2$); other anions: fluoride (F), and chloride (Cl); nutrients: phosphorus (P), phosphate ($PO_4$), and ammonium ($NH_4$); physical parameters: pH, and temperature; carbon: carbon dioxide ($CO_2$), total inorganic carbon (TIC), dissolved inorganic carbon (DIC), total organic carbon (TOC), and dissolved organic carbon (DOC). The included sample fractions are unfiltered, filtered, and extracted (that is, acid digested). We screened out sites identified as confidential or with other publication restrictions. A visual representation of the data processing completed during the preparation of SWatCh is presented in Fig. 1.

### 2.2.1 Flagging of Low-Quality Data

We identified low quality data using the flag "Rejected"; for example, samples flagged as "unreliable", "suspect", or "poor quality" in the source databases. Additionally, we flagged values below zero for all variables except temperature, alkalinity, and ANC; these values are assumed to have been entered incorrectly. A total of 79,910 data points were considered to have low data quality, representing 1.48% of data in SWatCh.

### 2.2.2 Removal of Duplicates

We removed duplicate site and sample data. Three of our source databases, GEMStat, the Global River Chemistry Database (GloRiCh), and Waterbase are compilations of water chemistry data from several sources, and thus repeat some measurements. We removed duplicated sites based on the unique site identification code. We removed duplicated samples based on the site identification code, date, variable name, variable fraction, variable speciation, and sample value. We define "variable fraction" as the component part of a water sample, such as filtered or unfiltered. "Variable speciation" is defined as the speciation of a reported parameter; for example, $NH_4$ may be reported as nitrogen (N) or $NH_4$.

### 2.2.3 Flagging of Potential Outliers

We identify potential outliers for each timeseries in SWatCh using a four-times median absolute deviation (MAD) cut-off value. The MAD is preferred to other methods of outlier removal when the data have a skewed distribution or large outliers are present (Leys et al., 2013; Rousseeuw & Hubert, 2011), as is common in water chemistry data (for example, Rotteveel & Sterling, 2020). Water chemistry data in SWatCh have a skewed distribution for most variables; thus, the MAD is suitable outlier screening approach. The equation for the MAD is presented in Eq. (1).

$$MAD = \frac{1}{n}\sum_{i=1}^{n}|x_i - \tilde{x}| \; where \; i = 1, \dots, n$$

(1)

Here, MAD is median absolute deviation, $x_i$ an observation, and $\tilde{x}$ the median (Rousseeuw & Hubert, 2011). A total of 0.27% (13,309 values) were flagged as potential outliers.

## 2.3 Data Standardization

### 2.3.1 Database Format

The SWatCh database conforms to the DataStream Water Quality (DS-WQX) schema, a standardized data format which specifies the allowable elements and dataset structure. The DS-WQX schema is a simplified adaptation of the United States Environmental Protection Agency (US EPA) WQX schema. The US EPA WQX schema is an implementation of the Environmental Sampling, Analysis, and Results data standard which conforms to the National Environmental Information Exchange Network (EN) standards (US EPA, 2021). The complexity of the US EPA WQX schema presents a barrier to entry (DataStream Initiative, 2022), which is addressed by the simplified DS-WQX schema. The US EPA WQX schema is simplified compared to the DS-WQX schema in the following ways: most optional columns are removed, data are stored as one database as opposed to multiple relational datasets, column names are simplified, and date and time information is conformed to the ISO 8601 format to simplify parsing and ensure universal readability (DataStream Initiative, 2022). For further details, please see https://github.com/datastreamapp/schema.

We made two changes to the DS-WQX format: the minimum value for the "MonitoringLocationLatitude" field (that is, minimum allowable latitude value) was updated from 0 to -90 so that sites located in the southern hemisphere could be included; and "OTHER" was added as an allowable value to the "ResultAnalyticalMethodContext" field (that is, the context associated with the analysis identifier code; for example, the agency which published the analysis method specifications), so method information which was undefined in the DS-WQX schema could be included. For samples where the "ResultAnalyticalMethodContext" was specified as "OTHER", information on the analysis identifier code context is provided in the "ResultComment" column.

### 2.3.2 Variable Naming and Measurement Units

We standardized variable naming conventions in accordance with DS-WQX. Variable names are indicated separately from variable fractions and speciation to facilitate analysis of different fractions simultaneously. In the input datasets, the fractions are not specified for all variables; for these, we denote the fraction as "Unspecified".

We harmonized the measurement units and variable speciation for each parameter to simplify data analysis. Measurements were reported in different units in the input datasets; we standardized them to the most common International System of Unit (SI unit) we observed for each variable. For example, Ca was reported in µg L$^{-1}$, mg L$^{-1}$, eq L$^{-1}$, and Mol, but was most commonly reported as mg L$^{-1}$, thus, we standardized the measurement unit to mg L$^{-1}$. Concentrations are provided in mg L$^{-1}$, other than Al and Fe (µg L$^{-1}$), ANC (mmol L$^{-1}$), pH (no unit: "None"), and temperature (°C).

Several input datasets did not include their encoding type, causing corrupted characters and measurement unit ambiguity. To prevent these errors, we omit non-ASCII (American Standard Code for Information Interchange) characters; for example, micrograms (µg), are denoted as ug. Measurement units in SWatCh conform to the DS-WQX standard.

### 2.3.3 Censored Data Notation

We standardized censored data notation to facilitate easier handling of these values. Censored data notation varied across the input datasets and included abbreviations such as "bdl", "<", or the numeric value of the detection limit. The input datasets did not distinguish between samples measured at or below the detection limit. Detection limits differed across and within datasets; thus, we standardized below detection limit values by flagging them and providing the detection limit in separate columns, allowing for various approaches of handling these results.

## 2.4 Mapping

We harmonized the coordinate reference systems (CRSs) of the sample site locations to simplify geographic analysis. Site location coordinates are provided in various CRSs in the input datasets; thus, we re-projected them to the World Geodetic System 1984 (WGS 84) geographic CRS. We selected WGS 84, as this provides good mean solution across the globe and can easily be projected to local datums (Bajjali, 2018).

## 3 Results

The SWatCh database contains water chemistry data across 24 variables, four fractions, 33,722 sites, and 5,062,980 samples collected between 1960 and 2022 (Table 2). SWatCh is available on Zenodo (DOI: 10.5281/zenodo.6484939; Rotteveel and Heubach, 2021). Sample collection frequency ranges from approximately twice a day to one-time samples, depending on the parameter and water body type. The parameters with the highest average annual sampling frequency are temperature (791) and pH (359). Average annual sampling frequency across all parameters is similar between water body types: four samples per year for lacustrine and riverine systems, and eight samples per year for reservoirs. Not all samples included collection and analysis methodologies; for the samples where this information was available, there are 565 different methods.
Sites in SWatCh are located across the globe, but are concentrated in North America, South America, and Europe (Fig. 2), and encompass a variety of bedrock types (United States Geological Survey, n.d.), land use types (Goldewijk et al., 2011), and climate zones (Kottek et al., 2006). The spatial distribution of sampling locations varies by water body type; notably, only riverine sites are available in northern North America and several island nations, such as Guam and New Zealand, and reservoir sites are concentrated in the equatorial and arid climate zones, such as Central America (Fig. 2).
The number of sites available to study freshwater acidification, and their spatial extent, decreases with number of included parameters and timeseries length (Table 3; Fig. 3). We allocated the available parameters in SWatCh into five groupings, each of which allows freshwater acidification to be studied

with increasing detail and certitude. The parameter groupings, in order of decreasing importance, are as follows:

1. Acidity (pH): pH is assigned the to the first grouping because it is the primary indicator of freshwater acidification.
2. Basicity (alkalinity, hardness, ANC, $CO_3$, $HCO_3$, Ca, and Mg): the primary measures of basicity are assigned to the second grouping because they are also used as primary indicators of freshwater acidification and can be used to determine the stage of freshwater acidification. For example, Stage 2 is characterized by an increase in freshwater $C_B$ concentrations as cation exchange from cation exchange sites in soils buffers acid anion deposition (Galloway et al., 1983). K and Na are not included in the basicity grouping because they are usually present in minor concentrations compared to Ca and Mg (Maybeck, 2004).
3. Acid anions ($SO_4$, $NO_3$, and $NO_2$): acid anions are assigned to the third grouping because acid deposition is usually the primary driver of freshwater acidification (Galloway et al., 1983), but may not be a good indicator of freshwater acidification in low $C_B$ waters, or waters with high DOC concentrations (Rotteveel & Sterling, 2020).
4. Metallic cations (Al and Fe): metallic cations are assigned to the fourth group because soils undergo Al or Fe buffering in response to acid deposition once soil $C_B$ are depleted (Björnerås et al., 2017; Galloway et al., 1983), and are thus a secondary indicator of freshwater acidification.
5. Weak acids ($CO_2$, TOC/DOC, and $NH_4$): although weak acids are not a primary driver of freshwater acidification in most catchments, they are an important driver of freshwater response to acid deposition in some catchments with low buffering capacity (Clair et al., 2011; Rotteveel & Sterling, 2020); thus, they are assigned to the fifth grouping.
6. Other (temperature, K, Na, Cl, F, P, $PO_4$, and DIC): the remaining parameters are assigned to the last group because they characterize catchment-scale processes which may have secondary effects on freshwater response to acid deposition (for example, Berger et al., 2015; Harriman et al., 1995; Kopáček et al., 2001). DIC is included in this group because the speciation (that is, $CO_2$, $CO_3$, or $HCO_3$) is unknown, although it can be calculated using pH.

## 4 Discussion

Here, we discuss the main limitations we encounter when compiling and analyzing datasets and provide recommendations for data sharing to facilitate more large-sample and global scale water chemistry research.

## 4.1 Data Availability and Spatial Gaps

Some variables have smaller sample sizes. The number of reported measurements differs greatly per variable, with metals (Fe and Al) and F having the smallest sample sizes and lowest sampling frequencies and pH and temperature having the largest and highest. This discrepancy is possibly due to these parameters being relevant to a wider range of research topics or the cost of measurement, where pH and temperature can be measured with a variety of field or laboratory-based multiparameter probes, whereas metals and anions require laboratory analysis. What is currently unknown, is if analysis results are

under-reported for some variables; that is, if all laboratory analysis results are reported for each sample included in the input databases. Prior research on one of the main variables with low sample size (Fe), includes an openly available research dataset of 340 water bodies in Europe and eastern North America (Björnerås et al., 2017). Despite the geographical coverage and size of this dataset, it is not included in SWatCh because the data do not adhere to the DS-WQX data schema due to missing variable fractionation information. These types of published research datasets are uncommon (Alsheikh-Ali et al., 2011) and highlight the potential contribution of unpublished raw research data.

Critical data gaps exist across large areas on the African, Asian, Australian, and Antarctic continents, representing mainly the equatorial, arid, snowy, and polar climate zones (Kottek et al., 2006). The zones of missing data represent regions where freshwater acidification is an emerging issue, for example in China (Li et al., 2019), and regions where climate change induced alteration of freshwater discharge regimes is projected the greatest by 2050 (Döll and Zhang, 2010). The lower data coverage in some of these regions represents a limitation in the development of global water chemistry models (Harrison, Caraco, et al., 2005; Harrison, Seitzinger, et al., 2005), and may inhibit the detection – and therefore treatment – of emerging water quality problems related to climate change induced perturbation of freshwater discharge regimes. The observed lower data availability may be because of our reliance on English datasets, less data sharing in these regions due to concerns about "parachute research" (where researchers abscond with local data to their home countries) (Serwadda et al., 2018), a lack of funding for scientific research (Serwadda et al., 2018), a lack of national data sharing regulations (Serwadda et al., 2018; Thu and Wehn, 2016), or outdated information management systems (Thu and Wehn, 2016). Despite the aforementioned data gaps, some of the most acidified regions of the world can be studied with a high degree of detail and certitude using SWatCh. Sites with sufficient available parameters (as defined in Results) and timeseries length (that is, ten to 15 years) to study drivers and trends are concentrated in the northern hemisphere, and encompass the some of the most acidified regions of North America and Europe (Björnerås et al., 2017; Clair, 2012; Clair et al., 2011; Driscoll et al., 2016) (Fig. 3). We chose a timeseries length of ten to 15 years because this is the minimum duration required to distinguish between short-term hydrological variability and underlying system behavior (Howden et al., 2011), and is comparable to timeseries lengths commonly used to study freshwater acidification (for example, Burns et al., 2008; Clair et al., 2011; Driscoll et al., 2016). Based on the available parameters for locations with a 10- or 15-year timeseries, the following aspects of freshwater acidification can be studied using SWatCh: acidification stage, extent of base cation depletion, catchment buffering processes, the importance of natural and/or weak acids, and other influential catchment-scale processes. The lack of water chemistry data relevant to freshwater acidification in some regions may be related to historical preferential research focus. That is, freshwater acidification research has historically predominantly been focused on Europe and North America (for example, Björnerås et al., 2017; Holland et al., 2005; Stoddard et al., 1999) where this is an established environmental issue, and less focused on other regions such as China, where this is an emerging concern (for example, Li et al., 2019).

Alleviating the issue of data availability is complex (Serwadda et al., 2018), but can be facilitated through journals more consistently implementing and enforcing data sharing policies (Alsheikh-Ali et al., 2011), ensuring coherence of and balance between data sharing policies and protecting national interests (Thu and Wehn, 2016), and engaging and crediting the people and organizations collecting the data (Serwadda et al., 2018).

## 4.2 Methodology Changes and Dissimilarity

The analysis of timeseries and intercomparison of data collected at different sites is challenging due to dissimilarity of sample collection programs and methodology changes. Methodology changes throughout a timeseries may result in spurious trend test results. For example, at site AL05BE0013, located in the Bow River approximately 4.5 km upstream of Canmore, Alberta, Canada, dissolved Al was analyzed using Variable Method Variable (VMV) 100195 prior to 2003, and VMV methods 107941 and
97963 after 2003 (Fig. 4). VMV methods 107941 and 97963 both use inductively coupled plasma mass spectrometry (ICP-MS) and have comparable low-level detection limits, whereas VMV 100195 uses inductively coupled argon plasma emission spectroscopy (ICAP) and has a higher detection limit. Because most values in this timeseries are lower than the detection limit for VMV 100195 (that is, 20 $\mu g \cdot L^{-1}$), analysis of the timeseries without removing samples analyzed via VMV 100195 would result in
the detection of a spurious negative trend. Similarly, disparate analysis methods across geographic regions may hinder comparability and consolidation of data collected by different sources (WHO and UNCF, 2017). For example, in the USA, Al samples may be analyzed by US EPA method 200.7, with an estimated detection limit of 45 $\mu g\ L^{-1}$ (US EPA, 2015), whereas in Europe, Al samples may be analyzed by ISO method 15586:2003, with an estimated detection limit of 1 $\mu g\ L^{-1}$ (ISO/TC 147 SC2,
2003); samples analyzed by these two methods cannot be compared if Al concentrations below are 45 $\mu g\ L^{-1}$. Trend analysis can also not be robustly performed if different sample fractions are present throughout the timeseries. For example, Environment and Climate Change Canada (ECCC) analyzed the unfiltered Al fraction as extractable Al ($Al_{ext}$; comprising the dissolved fraction and weakly bound or sorbed molecules) prior to 2011 in Atlantic Canada and as total Al ($Al_t$: comprising dissolved,
weakly bound or sorbed, and particulate molecules) after 2011 (Rotteveel & Sterling, 2020). To facilitate intercomparison of data and trend analysis, the creation of internationally standardized variable definitions and cross-boundary analysis methodology is needed (WHO and UNCF, 2017).

## 4.3 Ambiguity and Inconsistency

We encounter ambiguity and inconsistency in variable and faction naming conventions, reporting units,
analysis methodology, and dataset encoding. Firstly, we find variable and fraction definitions and consistency to be lacking in most input datasets. For example, an $Al_d$ sample may be filtered through a 0.45 or 0.10 $\mu m$ filter; both samples are considered $Al_d$ but represent a different set of Al molecules. Since naming conventions are variable, and there are no internationally standardized variable definitions (WHO and UNCF, 2017), defining variables and their fractions is required to prevent confusion regard-
ing comparability. Similarly, reporting units and censored data notation should be defined and consistent throughout the dataset; this includes spelling, abbreviations, and capitalization. We also observe ambiguity regarding analysis methodology, where analysis methods are inadequately described or missing entirely. Ideally, analysis method reporting includes all the following which are applicable: filter size and type, analysis instrument, acid preservative type, location of acid preservation (in field or la-
boratory), and the analysis/speciation method, method code, its publishing agency, and link to a reference document. Lastly, we encounter corrupted characters due to unknown dataset encoding; to prevent

this ambiguity, the encoding of the dataset should be known and published, this is especially important for datasets not encoded in 8-bit Unicode (UTF-8), which preferred for data exchange (ISO/IEC JTC 1/SC 2, 2017).

## 4.4 Limitations and Future Work

In addition to the challenges noted above, the main limitations of SWatCh are a lack of discharge data and information on watershed land use and land cover. We did not include discharge information, as there are numerous openly available global scale river discharge datasets which cover some of the sites available in the SWatCh database. For example, those available via the European Environmental Agency's Waterbase or the Global Runoff Data Centre. Further development is needed to integrate existing discharge datasets into SWatCh, allowing discharge-weighted water chemistry concentrations to be computed. The DS-WQX schema does not allow for the inclusion of watershed information such as land use and land cover; thus, we do not include these data in SWatCh. Some of this information is available in the input datasets, for example, the GloRiCh database (Hartmann et al., 2014). Catchment characteristics can be identified for sites by using existing global datasets such as HydroATLAS, which provides information on hydrologic, physiographic, climate, land use and land cover, soils and geology, and anthropogenic influences for catchments at a resolution of up to 15 arc-seconds (approximately 463 m at the equator) (Linke et al., 2019).

## 5 Conclusion

Prior research demonstrates that despite variability in sample size, geographic coverage, and analysis methodology, large-sample datasets facilitate the understanding of global water chemistry processes and the identification of transboundary problems (for example, Björnerås et al., 2017; Monteith et al., 2007; Weyhenmeyer et al., 2019). Despite these clear benefits, there are few global scale water chemistry datasets. We created SWatCh to begin to fill this gap; it is a global database of surface water chemistry focused on freshwater acidification-related variables. This database contains water chemistry data across 24 variables, four variable fractions, 33,722 sites, and 5,062,980 unique samples collected between 1960 and 2022. The numerous available variables and large sample sizes in SWatCh allows users to conduct powerful and robust statistical analyses to answer emerging global surface water chemistry questions. To facilitate data use in databases like SWatCh and by other researchers, we recommend making research data openly available, standardizing analysis methodology, and avoiding ambiguity/inconsistency in variable and fraction names, reporting units, censored data notation, analysis method descriptions, and dataset encoding. Future work should focus on filling the spatial data gaps identified in Asia, Africa, and Australia, and adding discharge data. With more people experiencing decreased water quantity (Burek et al., 2016; Mekonnen and Hoekstra, 2016), maintaining water quality is paramount. By facilitating the global exchange of their data, researchers can contribute toward this goal.

## Data Availability

The SWatCh database is available on Zenodo (DOI: 10.5281/zenodo.6484939); it can be accessed by navigating to https://zenodo.org/ and searching for dataset number 6484939. No account or sign-up is required to download the data. SWatCh is composed of third-party data, as listed in Table 1. GEMStat
data, 7,401 sites (21.95 % of sites), are not available in SWatCh due to a publication ban (Supplement S1). Users may add these data by requesting the GEMStat dataset from the United Nations Environment Programme and running the SWatCh data processing scripts available from the GitHub repository indicated below.

## Code Availability

The code used to generate the SWatCh database is published on GitHub:  https://github.com/Lob-keRotteveel/SWatCh.

## Author Contribution

LR conceived the original idea, compiled, and prepared the data, co-developed the data processing scripts, conducted the geospatial information systems (GIS) procedures, conceptualized and prepared
the figures and tables, and was the principal author. FH wrote the data validation scripts and co-developed the data processing scripts. SMS provided supervision and co-edited the manuscript.

## Competing Interests

The authors declare that they have no conflict of interest.

## Disclaimer

While substantial efforts are made to eliminate errors from the SWatCh database, complete accuracy of the data and metadata cannot be guaranteed. All data and metadata are made available "as is". Neither Lobke Rotteveel, Franz Heubach, and Dr. Shannon M. Sterling nor their current or future affiliated institutions, including the Sterling Hydrology Research Group and Dalhousie University, can be held responsible for harms, damages, or other consequences resulting from the use or interpretation of infor-
mation contained within the SWatCh database.

## Acknowledgements

Thank you to the United Nations Environment Programme and International Centre for Water Resources and Global Change, Environment and Climate Change Canada, the McMurdo Dry Valleys Long Term Ecological Research Team, United States of America National Science Foundation and National Water Quality Monitoring Council, the European Environment Agency, and Jens Hartmann, Ronny Lauerwald, and Nils Moosdorf for making the data collected by their contributing agencies, laboratories, researchers, and technicians openly available data for research. Thank you to Dr. Rob Jamieson for his feedback on the draft of this manuscript. Thank you to Abby Millard and Lilian Barraclough for assistance with compiling site data. This research was funded by the Nova Scotia Government through the Nova Scotia Graduate Scholarship program.

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

**Figure 1: Workflow for creating SWatCh. Below detection limit is abbreviated as BDL and coordinate reference system is abbreviated as CRS.**

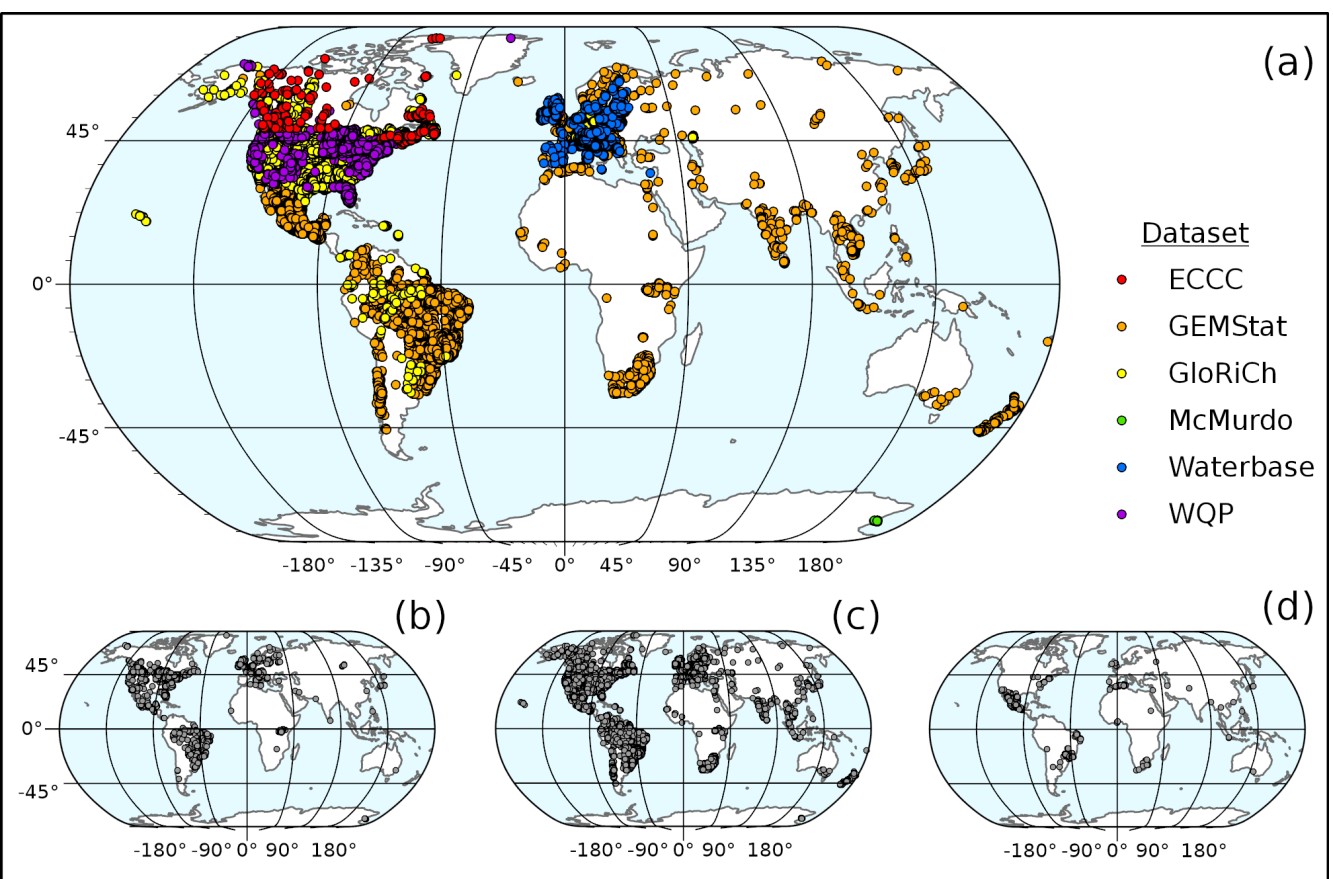

**Figure 2: Sample sites in the SWatCh database, colored by dataset source (a), and separated by site type: lakes/ponds (b), rivers/streams/canals (c), and reservoirs (d). Points overlap where sites are in close vicinity. Projection: Natural Earth, scales: 1:275,000,000 (a) and 1:725,000,000 (b, c, and d). Dataset abbreviations are defined in Table 1.**


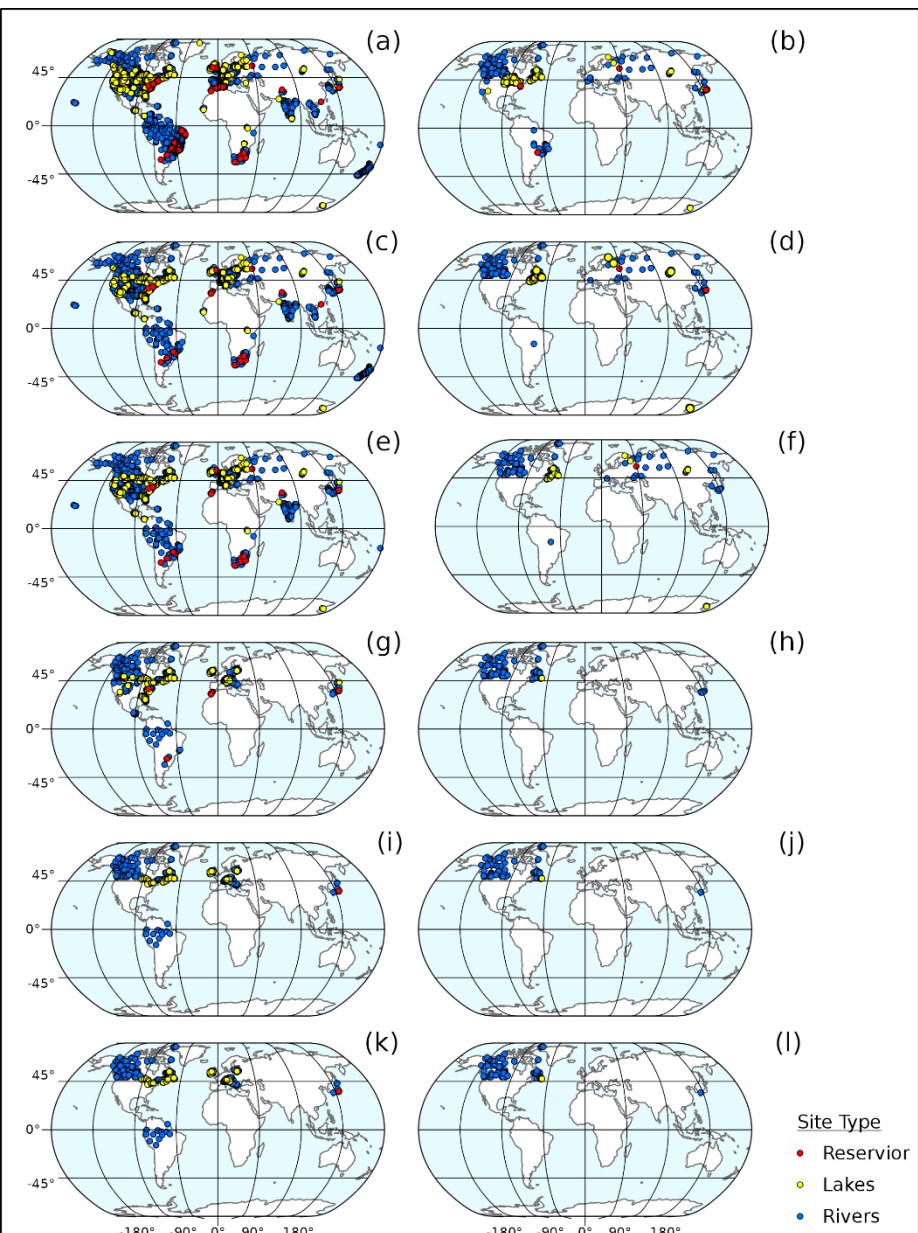

Figure 3: Sample size (sites) for waterbody types based on timeseries duration and data availability in the following parameter categories: acidity (a and b), basicity (c and d), acid anions (e and f), metallic cations (g and h), weak acids (i and j), and other parameters (k and l). Sites included in consecutive parameter categories also meet all the prior category requirements. Sites with at least one observation are shown in the left panes, and sites with a minimum timeseries length of 15 years are shown on in the right panes. Points overlap where sites are in close vicinity. Projection: Natural Earth, scales: 1:550,000,000.

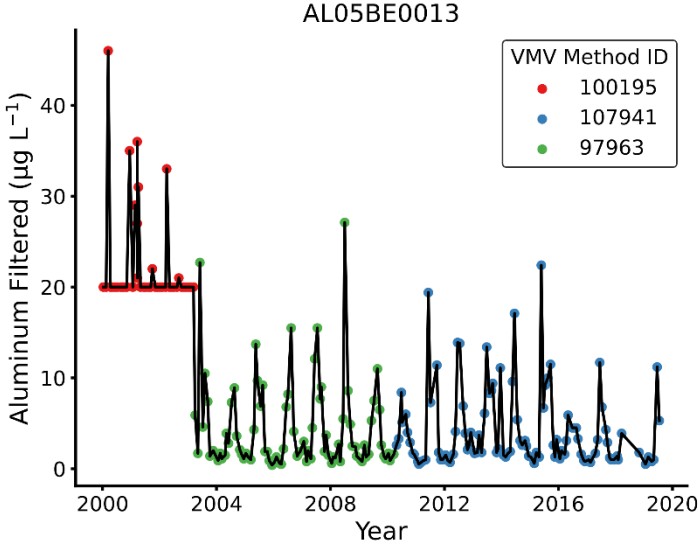


**Figure 4: Example of change in analysis methodology on detected concentrations. Colored points represent different analysis methodologies. For non-detect concentrations, the detection limit is shown.**

**Table 1: Data sources.**

| Dataset/Database | Source |
|---|---|
| Global Water Quality database and information system (GEMStat) | United Nations Environment Programme (2017). GEMStat database of the Global Environment Monitoring System for freshwater (GEMS/Water) Programme. International Centre for Water Resources and Global Change, Koblenz. Accessed 24 March 2022. Available upon request from GEMS/Water Data Centre: gemstat.org |
| Global River Chemistry Database (GloRiCh) | Hartmann, J., Lauerwald, R., Moosdorf, N. (2019). GLORICH - Global river chemistry database. PANGAEA. Accessed 18 August 2019. Available from: https://doi.org/10.1594/PANGAEA.902360. Supplement to: Hartmann, J. et al. (2014). A Brief Overview of the GLObal RIver Chemistry Database, GLORICH. Procedia Earth and Planetary Science, 10, 23-27, https://doi.org/10.1016/j.proeps.2014.08.005. |
| National Long-Term Water Quality Monitoring Database (ECCC) | Environment and Climate Change Canada (2019). National Long-term Water Quality Monitoring Data. Accessed 19 March 2022. Available from: http://data.ec.gc.ca/data/substances/monitor/national-long-term-water-quality-monitoring-data/ |
| Water Quality Database (WQP) | National Water Quality Monitoring Council (2019). Water Quality Portal. Accessed 7 September 2019. Available from: https://www.waterqualitydata.us. |
| Waterbase | European Environment Agency - European Environment Information and Observation Network (Eionet) (2019). Waterbase - Water Quality ICM. Accessed 5 April 2022. https://www.eea.europa.eu/data-and-maps/data/waterbase-water-quality-icm-1. |
| McMurdo Dry Valleys Long Term Ecological Research Network (McMurdo) | Gooseff, M.N., Lyons, W. (2022). Dissolved organic carbon (DOC) concentrations in glacial meltwater streams, McMurdo Dry Valleys, Antarctica (1990-2020, ongoing). Environmental Data Initiative. Accessed 4 April 2022. doi: 10.6073/pasta/878eccb6e5c8e492f933381b8c257d79. |
| | Gooseff, M.N., Lyons, W. (2022). Ion concentrations in glacial meltwater streams, McMurdo Dry Valleys, Antarctica (1993-2020, ongoing). Environmental Data Initiative. Accessed 4 April 2022. doi: 10.6073/pasta/275ee580f3c93f077dd7ddcce1f2ecdd. |

| Dataset/Database | Source |
|---|---|
| McMurdo Dry Valleys Long Term Ecological Research Network (McMurdo) | Gooseff, M.N., Lyons, W. (2022). Nitrogen and phosphorus concentrations in glacial meltwater streams, McMurdo Dry Valleys, Antarctica (1993-2020, ongoing). Environmental Data Initiative. Accessed 4 April 2022. doi: 10.6073/pasta/f6131f5ef67901bc98027e9df55ec364. |
| | Lyons, W. (2015). Dissolved Inorganic Carbon in Streams. Environmental Data Initiative. Accessed 4 April (2022). doi: 10.6073/pasta/4d64208bd91fc6a336c9c388436b1634. |
| | Lyons, W. (2015). Stream Nutrients for Reactivated Channel. Environmental Data Initiative. Accessed 4 April (2022). doi: 10.6073/pasta/b3d212996e5e4cb7f91b82090b4f550d. |
| | Lyons, W., Mcknight, D.M. (2015). Stream Chemistry for Reactivated Channel. Environmental Data Initiative. Accessed 4 April 2022. doi: 10.6073/pasta/ed143e49e82d0aaa1494447ebcee17c1. |
| | Priscu, J. (2018). Dissolved inorganic carbon (DIC) concentrations in discrete water column samples collected from lakes in the McMurdo Dry Valleys, Antarctica (1993-2017, ongoing). Environmental Data Initiative. Accessed 4 April 2022. doi: 10.6073/pasta/e68682ea6614259b4f091be206a773b8. |
| | Priscu, J. (2019). Hydrogen ion concentrations (pH) in discrete water column samples collected from lakes in the McMurdo Dry Valleys, Antarctica (1993-2018, ongoing). Environmental Data Initiative. Accessed 4 April 2022. doi: 10.6073/pasta/a0c17e313c63f6b5e5e5e071e5ba6b4a. |
| | Priscu, J. (2022). Dissolved organic carbon (DOC) concentrations in discrete water column samples collected from lakes in the McMurdo Dry Valleys, Antarctica (1993-2022, ongoing). Environmental Data Initiative. Accessed 4 April 2022. doi: 10.6073/pasta/a5d82d5d2167679c8ecff0d8ad06c0ee |
| | Priscu, J. (2022). Nitrogen and phosphorus concentrations in discrete water column samples collected from lakes in the McMurdo Dry Valleys, Antarctica (1993-2020, ongoing). Environmental Data Initiative. Accessed 4 April 2022. doi: 10.6073/pasta/5cba7e25aa687c1e989c72c3ee0a0f69. Dataset accessed 4 April 2022. |
| | Priscu, J., Welch, K.A., Lyons, W. (2022). Ion concentrations in discrete water column samples collected from lakes in the McMurdo Dry Valleys, Antarctica (1991-2019, ongoing). Environmental Data Initiative. Accessed 4 April 2022. doi: 10.6073/pasta/31f7354d1a05679eb3ce7c384c6e2b22. |


**Table 2: Summary of sample size, timeseries start and end dates, and average annual sampling frequency throughout timeseries separated by water body type and variable. Minimum and maximum are abbreviated as min. and max., respectively.**

| Site Type | Parameter | Sample Size | | Earliest Data Point (yr) | | | Latest Data Point (yr) | | | Average Annual Sampling Frequency | | |
|---|---|---|---|---|---|---|---|---|---|---|---|---|
| | | Sites | Samples | Min. | Max. | Mean | Min. | Max. | Mean | Min. | Max. | Median |
| Lake/ Pond | ANC | 285 | 5,151 | 2013 | 2019 | 2015 | 2013 | 2020 | 2016 | 1 | 74 | 4 |
| | Al | 472 | 5,553 | 2000 | 2018 | 2013 | 2000 | 2019 | 2016 | 1 | 29 | 2 |
| | Alkalinity | 210 | 12,317 | 1977 | 2019 | 2001 | 1982 | 2020 | 2010 | 1 | 34 | 6 |
| | $CO_2$ | 2 | 53 | 1979 | 1980 | 1980 | 1980 | 1985 | 1983 | 1 | 9 | 5 |
| | Ca | 2,592 | 24,385 | 1993 | 2019 | 2012 | 1995 | 2020 | 2013 | 1 | 73 | 2 |
| | Cl | 3,105 | 67,153 | 1993 | 2019 | 2010 | 1995 | 2020 | 2013 | 1 | 107 | 4 |
| | F | 491 | 3,976 | 1994 | 2019 | 2014 | 2002 | 2019 | 2015 | 1 | 21 | 2 |
| | Fe | 266 | 6,282 | 2000 | 2019 | 2012 | 2000 | 2019 | 2017 | 1 | 23 | 7 |
| | $HCO_3$ | 520 | 14,222 | 1969 | 2018 | 2014 | 1974 | 2020 | 2018 | 1 | 63 | 6 |
| | Hardness | 508 | 12,141 | 1990 | 2019 | 2012 | 1996 | 2019 | 2017 | 1 | 80 | 2 |
| | TIC/DIC | 65 | 1,384 | 1993 | 2010 | 2001 | 1995 | 2017 | 2009 | 1 | 4 | 2 |
| | K | 1,315 | 15,551 | 1993 | 2019 | 2010 | 1995 | 2020 | 2012 | 1 | 50 | 1 |
| | Mg | 1,923 | 20,467 | 1993 | 2019 | 2011 | 1995 | 2020 | 2012 | 1 | 70 | 1 |
| | $NH_4$ | 1,037 | 32,022 | 1993 | 2019 | 2014 | 2003 | 2020 | 2017 | 1 | 79 | 6 |
| | $NO_2$ | 1,176 | 25,152 | 1993 | 2020 | 2012 | 2000 | 2020 | 2015 | 1 | 75 | 4 |
| | $NO_3$ | 1,502 | 30,696 | 1993 | 2020 | 2012 | 2000 | 2020 | 2014 | 1 | 122 | 4 |
| | Na | 1,694 | 18,865 | 1993 | 2019 | 2010 | 1995 | 2020 | 2012 | 1 | 62 | 2 |
| | TOC/DOC | 529 | 15,032 | 1993 | 2019 | 2012 | 2000 | 2022 | 2016 | 1 | 52 | 5 |
| | P | 8,384 | 227,921 | 2000 | 2020 | 2009 | 2000 | 2020 | 2013 | 1 | 105 | 4 |
| | $PO_4$ | 1,272 | 23,537 | 2000 | 2019 | 2012 | 2000 | 2020 | 2014 | 1 | 33 | 4 |
| | $SO_4$ | 2,616 | 22,553 | 1993 | 2019 | 2011 | 1995 | 2020 | 2013 | 1 | 80 | 1 |
| | Temperature | 8,302 | 1,053,822 | 2000 | 2020 | 2009 | 2000 | 2020 | 2012 | 1 | 791 | 5 |
| | pH | 6,490 | 566,977 | 1993 | 2019 | 2009 | 1994 | 2020 | 2012 | 1 | 359 | 6 |
| Reservoir | Al | 9 | 301 | 2000 | 2014 | 2005 | 2004 | 2015 | 2010 | 1 | 9 | 5 |
| | Alkalinity | 44 | 9,836 | 1976 | 2016 | 1986 | 1981 | 2020 | 2001 | 1 | 159 | 9 |
| | $CO_2$ | 1 | 79 | 1980 | 1980 | 1980 | 1995 | 1995 | 1995 | 7 | 7 | 7 |
| | Ca | 30 | 4,091 | 2000 | 2015 | 2006 | 2002 | 2018 | 2013 | 1 | 155 | 7 |
| | Cl | 49 | 4,013 | 2000 | 2014 | 2001 | 2000 | 2018 | 2011 | 1 | 15 | 5 |
| | F | 13 | 2,928 | 2000 | 2006 | 2000 | 2002 | 2012 | 2009 | 4 | 123 | 11 |
| | $HCO_3$ | 2 | 101 | 2016 | 2016 | 2016 | 2017 | 2017 | 2017 | 24 | 27 | 25 |
| | Hardness | 600 | 8,634 | 1976 | 2019 | 2013 | 1996 | 2019 | 2018 | 1 | 60 | 2 |
| | K | 27 | 3,771 | 2000 | 2015 | 2007 | 2000 | 2018 | 2013 | 1 | 148 | 8 |
| | Mg | 30 | 3,837 | 2000 | 2015 | 2006 | 2002 | 2018 | 2013 | 1 | 154 | 7 |
| | $NO_2$ | 107 | 6,789 | 2000 | 2015 | 2005 | 2004 | 2015 | 2012 | 1 | 24 | 6 |
| | $NO_3$ | 130 | 7,456 | 2000 | 2015 | 2005 | 2005 | 2017 | 2012 | 1 | 29 | 6 |
| | Na | 29 | 3,851 | 2000 | 2015 | 2006 | 2001 | 2018 | 2012 | 1 | 151 | 8 |
| | TOC/DOC | 1 | 255 | 2001 | 2001 | 2001 | 2015 | 2015 | 2015 | 18 | 18 | 18 |
| | P | 6 | 250 | 2000 | 2014 | 2009 | 2011 | 2015 | 2013 | 1 | 11 | 2 |
| | $PO_4$ | 52 | 5,747 | 2000 | 2009 | 2004 | 2008 | 2010 | 2010 | 5 | 133 | 10 |
| | $SO_4$ | 29 | 4,074 | 2000 | 2015 | 2006 | 2002 | 2018 | 2013 | 1 | 155 | 8 |
| Reservoir | Temperature | 161 | 9,753 | 2000 | 2015 | 2004 | 2000 | 2018 | 2012 | 1 | 35 | 6 |
| | pH | 210 | 15,688 | 2000 | 2014 | 2004 | 2000 | 2018 | 2012 | 1 | 155 | 7 |

| Site Type | Parameter | Sample Size | | Earliest Data Point (yr) | | | Latest Data Point (yr) | | | Average Annual Sampling Frequency | | |
|---|---|---|---|---|---|---|---|---|---|---|---|---|
| | | Sites | Samples | Min. | Max. | Mean | Min. | Max. | Mean | Min. | Max. | Median |
| River/ Stream/ Canal | ANC | 479 | 8,802 | 2013 | 2019 | 2015 | 2013 | 2019 | 2017 | 1 | 24 | 4 |
| | Al | 967 | 59,043 | 2000 | 2019 | 2010 | 2000 | 2019 | 2016 | 1 | 74 | 4 |
| | Alkalinity | 4,263 | 350,531 | 1960 | 2019 | 1994 | 1968 | 2020 | 2001 | 1 | 112 | 3 |
| | $CO_2$ | 65 | 6,966 | 1979 | 2007 | 2000 | 1981 | 2019 | 2017 | 1 | 33 | 4 |
| | $CO_3$ | 1,272 | 78,474 | 1960 | 2015 | 1979 | 1961 | 2020 | 1988 | 1 | 41 | 6 |
| | Ca | 2,495 | 72,758 | 1971 | 2020 | 2012 | 1971 | 2020 | 2015 | 1 | 52 | 4 |
| | Cl | 2,909 | 69,272 | 1972 | 2019 | 2012 | 1972 | 2020 | 2015 | 1 | 57 | 5 |
| | F | 2,925 | 62,930 | 1967 | 2019 | 1994 | 1967 | 2020 | 1999 | 1 | 44 | 2 |
| | Fe | 998 | 69,307 | 2000 | 2020 | 2012 | 2011 | 2020 | 2018 | 1 | 74 | 11 |
| | $HCO_3$ | 2,631 | 112,069 | 1960 | 2020 | 1989 | 1962 | 2020 | 1996 | 1 | 43 | 3 |
| | Hardness | 4,364 | 331,545 | 1960 | 2020 | 2010 | 1970 | 2020 | 2016 | 1 | 104 | 5 |
| | TIC/DIC | 353 | 18,266 | 1973 | 2016 | 1994 | 1974 | 2019 | 2002 | 1 | 41 | 3 |
| | K | 2,271 | 57,642 | 1972 | 2019 | 2009 | 1973 | 2020 | 2013 | 1 | 40 | 4 |
| | Mg | 2,414 | 72,945 | 1973 | 2020 | 2011 | 1973 | 2020 | 2014 | 1 | 52 | 4 |
| | $NH_4$ | 7,504 | 111,903 | 1971 | 2019 | 2003 | 1972 | 2020 | 2006 | 1 | 52 | 3 |
| | $NO_2$ | 8,477 | 145,325 | 1970 | 2020 | 2003 | 1973 | 2020 | 2006 | 1 | 52 | 3 |
| | $NO_3$ | 6,005 | 144,335 | 1986 | 2020 | 2009 | 1989 | 2020 | 2013 | 1 | 52 | 4 |
| | Na | 2,023 | 60,865 | 1975 | 2020 | 2011 | 1980 | 2020 | 2015 | 1 | 52 | 4 |
| | TOC/DOC | 2,943 | 91,423 | 1971 | 2019 | 2010 | 1971 | 2020 | 2013 | 1 | 81 | 5 |
| | P | 8,663 | 171,177 | 1970 | 2020 | 2007 | 1972 | 2020 | 2009 | 1 | 290 | 4 |
| | $PO_4$ | 7,393 | 107,069 | 1969 | 2019 | 2002 | 1972 | 2019 | 2005 | 1 | 39 | 2 |
| | $SO_4$ | 2,923 | 75,552 | 1970 | 2019 | 2010 | 1971 | 2020 | 2013 | 1 | 57 | 4 |
| | Temperature | 9,610 | 230,617 | 1982 | 2020 | 2009 | 1984 | 2020 | 2012 | 1 | 52 | 4 |
| | pH | 10,363 | 257,499 | 1980 | 2020 | 2008 | 1980 | 2020 | 2011 | 1 | 114 | 4 |


**Table 3: Sample size (sites) for waterbody types based on available parameters and timeseries duration. Sites included in consecutive parameter categories also meet all the prior category requirements.**

| Site Type | Category | | Minimum Timeseries Length (yr) | | | | |
|---|---|---|---|---|---|---|---|
| | | | 1 | 5 | 10 | 15 | 20 |
| River/ Stream/ Canal | 1. | Acidity | 10,363 | 2,712 | 1,013 | 237 | 2 |
| | 2. | Basicity | 5,026 | 964 | 325 | 208 | 1 |
| | 3. | Acid Anions | 4,029 | 801 | 291 | 184 | 1 |
| | 4. | Metallic Cations | 1,049 | 390 | 193 | 125 | 0 |
| | 5. | Weak Acids | 717 | 380 | 184 | 120 | 0 |
| | 6. | Other | 717 | 380 | 184 | 120 | 0 |
| Lake/ Pond | 1. | Acidity | 6,490 | 1,635 | 735 | 420 | 5 |
| | 2. | Basicity | 2,474 | 408 | 71 | 55 | 5 |
| | 3. | Acid Anions | 1,989 | 371 | 66 | 55 | 5 |
| | 4. | Metallic Cations | 447 | 73 | 44 | 1 | 0 |
| | 5. | Weak Acids | 242 | 73 | 44 | 1 | 0 |
| | 6. | Other | 242 | 73 | 44 | 1 | 0 |
| Reservoir | 1. | Acidity | 210 | 137 | 75 | 26 | 0 |
| | 2. | Basicity | 33 | 27 | 14 | 4 | 0 |
| | 3. | Acid Anions | 31 | 19 | 11 | 3 | 0 |
| | 4. | Metallic Cations | 5 | 3 | 2 | 0 | 0 |
| | 5. | Weak Acids | 1 | 1 | 0 | 0 | 0 |
| | 6. | Other | 1 | 1 | 0 | 0 | 0 |