# Peer review of "The Surface Water Chemistry (SWatCh) database: A standardized global database of water chemistry to facilitate large-sample hydrological research"

_Earth System Science Data, 2021_

## Author Comment (AC1)

**SWatCh Manuscript Review**

- Black test: reviewer comments
- Red text: responses

**Mary Kruk, 13 September 2021**

Hello,

With your manuscript currently under review I thought this would be a good opportunity to highlight a Canadian water quality database that currently addresses some of the challenges outlined in your paper.

I work on DataStream, an open access platform for sharing water quality data. It allows users to access, visualize, and download water quality datasets collected by monitoring programs across regional hubs in Canada (the Mackenzie Basin, Lake Winnipeg Basin, Atlantic Canada, and a Great Lakes hub coming October). A main focus of DataStream is to help community monitoring groups, citizen scientists, researchers, and governments share their data at a regional-scale by adopting the US EPA/USGS WQX data standard to promote data (re)use and interoperability in transboundary watersheds.

We thought it would be relevant to reach out because DataStream has faced many of the same challenges you address in your paper -- such as differing sample collection/analytical methods, reporting ambiguity, and spatial data gaps across Canada. We have found that the adoption of the WQX schema, used in the US Water Quality Portal, has helped us to align data collected by a wide range of monitoring initiatives. DataStream requires metadata on sample collection and analytical methods with each data point and reduces variable naming ambiguity by using the WQX list of allowed values for water chemistry parameters. We are constantly trying to evolve and improve the DataStream data standard and platform to better address these issues as I'm sure you are aware it is a large undertaking.

Given the alignment between your area of research and our work with DataStream I would encourage you to review the DataStream schema (https://github.com/gordonfn/schema) for consideration in your manuscript.

Sincerely,

Mary Kruk

Thank you very much for taking the time to review our manuscript and provide feedback. We enjoyed the off-line discussions regarding the points you raised. After reviewing the DS-WQX and US EPA/USGS WQX schema, we agree that the DS-WQX schema is highly suitable for SWatCh and will increase the inter-usablity of SWatCh with other existing large-sample datasets. Adapting SWatCh to conform to the DS-WQX schema also addresses several of Reviewer 1 and 2 comments, as discussed below. Based on these justifications, we will adapt SWatCh to conform to the DS-WQX schema.

**Anonymous Reviewer #1**

The manuscript by Rotteveel and Sterling presents the global surface water chemistry (SWatCh) database, which contains data for 17 variables (Al, Fe, major ions, nutients, organic C, pH, etc.) from 9 million samples collected between 1960 and 2019. This database has the specific purpose to support research on surface water acidification. To create this database, the authors used data from 6 exiting hydrochemical databases/dataset, which they put in a uniform format, and then removed samples that were flagged as problematic and duplicates that exist as some of the databases used have culled data from the other databases.

I was able to download and use SWatCh without any problem. The download process is straight forward, and the database is easy to use.

While it is a very important task to assemble available data into such a large, publically available database, I feel that the authors have done a very poor job with regard to quality checks. They only discarded data that was already flagged as problematic, or which had very clearly unrealistic values, which was limited to negative values for concentrations. I think a much more robust quality check would be required to publish this dataset with an article in ESSD. Further, I feel that the authors did a rather poor job at analysing and presenting the data. For these two points, please see my major comments further below. Finally, I would like to highlight that this database does not contain any data on alkalinity, acid neutralisation capacity (ANC), DIC or HCO3- concentrations. This information can be found in at least a few of the databases from which data was taken for SWatCh. More importantly, these parameters represent the buffering capacity of a surface water body against acidification, and would thus be of huge importance for the study of surface water acidification and recovery. It is completely incomprehensible for me why these parameters were not included in SwatCh.

I suggest that major revision are necessary before this study can be considered for publication in ESSD. Please, see my major and general comments below:

Thank you very much for your thorough review of our manuscript and your feedback regarding the content, quality, and presentation of the database. We have addressed your comments on database quality and presentation in the "Major Comments" below.

Agreed regarding your comment on included parameters. We recognize the importance of alkalinity, acid neutralization capacity (ANC), dissolved inorganic carbon (DIC), and bicarbonate ($HCO_3^-$) in determining surface water buffering capacity. Alkalinity and ANC are calculated parameters, and reported results may differ based on the calculation, for example, alkalinity may be expressed as total, hydroxide, carbonate, or bicarbonate alkalinity but reported in the input dataset as "alkalinity". Since sample fractionation and analysis/calculation methods were often omitted in the input datasets, we originally elected to omit these parameters due to the uncertainty associated with the reported values. Based on the feedback we have received, we will add the parameters you have listed (i.e., alkalinity, ANC, DIC, and  bicarbonate) in addition to carbonate ($CO_3^{2-}$) and partial pressure of carbon dioxide ($pCO_2$), when these parameters are available in the input datasets.

Major comment #1: Quality checks of database

You should check all parameter values if they are reasonable, even if they are not flagged. You should check for instance for unreasonable high values, which can be due to mistakes made with the units (in particular for a database like GloRiCh, where data was assembled from lots of different dataset). If for instance mg and ug (or mM and uM) have been mixed up at some point (that could already be a mistake in the dataset you are taking data from), this might lead to errors of three orders of magnitude. I would suggest to first define for each parameter a realistic value range. Then, for all values lying outside of that range, you should first check is that concerns only one value in a time series, or the whole time series of a sampling site, or all values of a certain data source (note that for instance GloRiCh gives references of the data sources it used, and already in GloRich such mistakes might be present). If extreme values concern one specific sampling location, it might be worth investigating if that might be due to an exceptional site. For instance extremely high F- concentrations might be due to hydrothermal influence. Extreme PO4- concentrations can be due to phosphate deposits, like in the Peace River catchment, Florida. Sediments from dried out lakes might yield high concentrations in NaSO4. Etc.

Agreed, we will improve the data validation in SwatCh in the following ways:

1. Data included in SWatCh will now be validated using the DS-WQX data schema, which is based on the US EPA/USGS WQX schema, an accepted data validation schema for similar databases established by an authoritative scientific body. The WQX schema does not include validation of realistic values for specific parameters, because, as you correctly state, the realistic ranges values are dependant on geographic and catchment-specific properties and thus difficult to define for such a geographically diverse dataset.

2. Data included in SWatCh will also undergo an additional data validation step beyond what is completed in the DS-WQX schema; values outside of previously-established globally-defined ranges (Maybeck, 2003) will be flagged, for parameters where this data is available.

As suggested by Reviewer #2, we now retain the potentially erroneous data in SWatCh and provide a flag identifying the data as such, as opposed to removing these data points.

For each sampling location you should look at the time-series and try to identify potential outliers within the time series. For each outlier, you might want to check if other parameters are also affected, which could mean that either something exceptional has happened or that data from another sampling location has been wrongly attributed. Anyway, you should flag those values. You cannot assume that all suspicious data has already been flagged accordingly, in particular as data comes from very different sources, and some of them, like GloRiCh, are again assembled from different sources with different degree of quality checks.

Concidering the size of the database and the number of included sites (38,598), this can realistically only be achieved programically. Thus, we would have to pre-define a statistically valid method of identifying outliers and remove/flag them based on those results.

Disagreed, it is beyond the reasonable scope of work robustly determine outliers in individual (and often multiple) timeseries for 38,598 sites, especially considering which values are defined as outliers is dependant on the statistical method used. This additional data cleaning step is beyond the scope of

work for SWatCh and is expected to be completed by researchers using the dataset as part of the standard data processing procedure prior to data analysis. This will be clarified in the database metadata and the manuscript.

Major comment #2: Presentation of database

Your results section is very short, and your discussion section doesn't make many links to your own results. Figure 2 is a good beginning to represent the available data, but it would be more interesting if the spatial coverage was represented separately for different types of inland water bodies. It is not clear at all from your manuscript how well lakes vs. reservoirs vs. rivers are represented in tat database.

Agreed, Figure 2 will be expanded to demonstrate the spatial data coverage by water body type.

It would also be interesting to know the numbers of samples per water body type that have measurements for a specific combination of parameters that are interesting with regard to acidification, like: How many samples are there with all major ions and pH? Here you should maybe start with an overview of which combinations of parameters are usually used to study acidification. I guess samples where only sodium or phosphate was measure are not that interesting. Maybe you can make a ranking of parameter combinations that allow you to study acidification with a different degree of conclusiveness and certitude. And then list the number of samples that have measurements for these parameter combinations, and do that separately for different kinds of water bodies (lakes, reservoirs, canals, ditches, rivers, etc.) and different world regions (at least continents, or major biomes/climate zones). You should also give an overview about which time-periods are covered in different parts of the world. That would very important if you want to investigate temporal trends in acidification recovery.

You should also think about presenting data density (number of sites, number of samples per site, average length and frequency of time-series, etc. ) for different types of inland waters as a map. You could take inspiration from figure 2a in Regnier et al. 2013 (Nature Geoscience,    DOI: 10.1038/ngeo1830), that created a density index for pCO2 values.

Agreed, additional tables and figures describing sample sizes and densities for differing sample parameter combinations, water body types, and timeseries frequency and length would be beneficial to the manuscript and will be added. Further discussion regarding the implications of sample size limitations in studying freshwater acidification at the global scale will also be added.

You should also take into account global geodatasets that allow for regional classification of water bodies, like for instance the HydroAtlas (Linke et al. 2019, https://doi.org/10.1038/s41597-019-0300-6). Like this you could make more qualified statement about which kind of river or lake is underrepresented in your dataset. You state you cannot link your chemistry data to catchment properties, but with HydroAtlas you could get a good idea what kind of river-catchment systems are well represented and what kind underrepresented.

Disagreed. We recognize that associating catchment characteristic information with water chemistry sample locations is possible; however, this is beyond the intended scope of SWatCh (i.e., a water chemistry database). By providing a shapefile of sample locations, SWatCh enables researchers to associate the water chemistry data with catchment characteristics currently already available in other published datasets, such as the one you recommend.

General comments:

L74-76: How did you perform those checks? Where can I see the results? I think a quality assessment of this kind is very important.

Acknowledged. This comment refers to "We assume that water chemistry data available from these reputable sources have undergone standard laboratory quality assurance and control; spot-checks of available methodology information support this assumption." The sample QAQC methodology implemented in SWatCh is in accordance with that used in other reputable large sample datasets such as the US EPA/USGS WQX database (formerly the STORET database). The spot-checks we refer to were a high-level review of the methodology used for sample analysis; for example, by verifying that an analysis method was approved by the publishing agency. Since sample certificates or analysis (COAs) are not available, additional verification of laboratory QAQC measures is not possible. We will remove the statement in question from the manuscript. We endeavour to further improve the data quality in SWatCh by implementing the additional data validation steps outlined in response to your Major Comment #1 above.

L85-86: I wonder how you identified these "untreated" water bodies. I know that in GloRiCh this information is not given. Here, some analysis of the water chemistry data itself could have been useful to spot suspicious cases, for which some investigation could have been performed based on the location information.

Acknowledged. As stated on line 85 of the manuscript, we define "untreated" as "water that is not wastewater or receiving treatment plant effluent near to the sample collection site." We will clarify this statement to indicate that we include only the following site types: river, stream, canal, lake, impoundment, or reservoir.

L88: By "phosphorus", do you mean "total phosphorus"?

Acknowledged. We include both total and dissolved phosphorus in the SWatCh dataset; hence, no fractionation is identified here; fractionation is shown in Table 3. We will add a definition of phosphorus and the other parameters included in SWatCh to Section 2.2 (Data Inclusion).

L91-92: Error message instead of reference.

Acknowledged, this error will be addressed in the manuscript revisions.

Section 4.2: When discussing these effect of methodological changes on time-series data, you should combine that indeed with an analysis of at least the longest time-series you have in your database.

Agreed, an example demonstrating the effects of methodological changed on timeseries interpretation will be included in Section 4.2.

Section 4.4: Here you mention that you often do not have the discharge data associated to the water chemistry data. Did you try to match the river water sampling locations with stream gauges from the Global Runoff Data Centre (GRDC, https://www.bafg.de/GRDC/EN/Home/homepage_node.html)?

Acknowledged. As you correctly state, discharge data may be available for some sites; there are approximately 10,000 stations included in the Global Runoff Data Centre compared to the

approximately 38,600 stations included in SWatCh. As stated above, the intended scope of SWatCh is limited to a water chemistry database; and inclusion of other data such as discharge is beyond the current scope of work. By providing a shapefile of sample locations and information on sample location IDs and sample collection date/times, SWatCh enables researchers to associate the water chemistry data with other types of information such as discharge.

**Anonymous Reviewer #2**

**Comments and Responses**

The authors present a newly created database on chemical composition of surface waters. The database is comprised of several database sources from which specific parameters/variables are extracted and unified for the specific purpose to provide a data base for surface water acidification research.

The collection and harmonization of data on water chemistry is very important to the research community, as it enables more refined global analyses of matter fluxes, temporal developments, climate change impacts, any many more.

The manuscript addresses an important data topic, which makes it worth to be published. However, due to the points stated below, I recommend a major revision.

Thank you for your detailed review of our manuscript. We appreciate your feedback regarding data quality, harmonization, and selection, database structure and presentation, and text quality. We have responded to your comments below.

Data quality

I would argue, from a personal viewpoint, that if the goal is to provide global coverage of data to enable global cross-boundary evaluation of surface waters, it may not be very important to have a high data quality, as the available amount of data will level out "outliers" or differences in the data analyses from a statistical viewpoint.

Agreed, we have endeavoured to strike the balance between your comments and those of Reviewer 1 by flagging values outside of previously-established globally-defined ranges (Maybeck, 2003), for parameters where this data is available.

Data harmonisation

The calls for a unified approach in all future data collections are very noble, but I doubt that they will be heard. Data producing authorities very often have their own, historically grown structures and formats, that are so convoluted and unpredictable that it would be and hopeless to expect a globally unified data structure

Acknowledged; no changes required.

Data selection

The authors state that the parameters were specifically selected to evaluate surface water acidification, however I would argue that the most important parameter in this regard is missing: total alkalinity

(TA). This is reported in some of the used sources, even if it may be in awkward units sometimes. The TA is fundamental for the understanding of the carbonate system and the interaction of CO2 and natural waters. Alternatively, dissolved inorganic carbon could be included, or both parameters, where available, to be able to calculate the missing parts of the carbonate system (TA and pH or DIC and pH enable the calculation of DIC or TA, respectively). Furthermore, the inclusion of TA would enable the calculation of a charge balance, which could provide an indicator for the data quality.

Agreed, please see our response to Reviewer #1 above, who raised the same issue.

Database structure and presentation

I really appreciate the approach of publishing the scripts for the database of Github. This makes the work very transparent and should be an example for all scientists working with complex data processing.

Thank you.

The chosen format of the data is slim and straightforward, however, for the average enduser, the relational style of the files may present a potential problem as data cannot be filtered and used as is, but have to be transformed. It may be an advantage (not a requirement) to provide a python script that converts the data into the "classical" column-row-format. It may, however, increase the filesize to an extent that makes it hard to handle.

Acknowledged. Based on the size of the three individual relational databases in SWatCh, the combined and expanded (i.e., column-row format) size of the datasets would likely be several hundred gigabytes. A file this size is too large to read into a spreadsheet program such as Excel without truncating the dataset; indicating that  it would remain unusable for the average end user. Although providing the data in relational databases complicates linking site and method information to sample analysis results, the datasets are currently small enough to read into a spreadsheet program without truncation, in which they can be filtered for the locations, sample parameters, date ranges, etc. of choice. Based on this, no change will be made to the database structure.

Regarding the units, the choice of weight units is okay but may lead to the need to recalculate to molar units as this is needed in geochemical calculations (e.g., charge balance, ratios, chemical formulas).

Acknowledged; no changes required.

Text quality

There are several typos, duplications and wording issues in the text. I mention some of them below. Overall, the text could benefit from a revision, which clears out the errors but more specifically narrows focus on the specific arguments for the need of a new and harmonized database.

Agreed. The text will be reviewed for errors prior to re-submission. By addressing your specific comments below regarding L47 below, we aim to clarify why a harmonized water chemistry database is required to answer global-scale surface water acidification research questions.

Specific comments

L8        2x "identify"

Acknowledged; text will be updated.

L18     Define the need for more data collection – how would that improve global models? Little data from arid regions may also be due to the fact that there are less surface waters

L19     "Environs"

Acknowledged; text will be updated.

L21/22   2x "address"

Acknowledged; text will be updated.

L29     "a number projected…" is meant to refer to the 4 bln people, but as it stands in the text it rather refers to "at least one month"

Agreed, text will be revised to state "With two-thirds of the global population (4.0 billion people) already experiencing water shortages at least one month per year (Mekonnen and Hoekstra, 2016), and 4.8-5.7 billion people projected to experience water shortages by 2050 (Burek et al., 2016)..."

L30     "these resources" – which?

Acknowledged, text will be revised to state "... maintaining the quality of drinking water sources is paramount to human health and society."

L36     Define "transboundary problem"

Acknowledged, in the context of this manuscript we define a "transboundary problem" to be a water quality issue, or cause of a water quality issue, which crosses international borders. For example, a main driver of freshwater acidification in Atlantic Canada is acid deposition originating from all the major production regions in North America, including those in the United States of America (Shaw, 1979). A similar definition of "transboundary problem" is often used when discussing water availability issues which cross international borders (e.g., Thu and Wehn, 2015). Definition will be added to the text.

L47     When I comes to the fate and behavior of compounds in natural water, I would argue, the catchment scale is a good and proven approach. I may not understand the term "transboundary" in your sense, but why should be look transboundary if fluxes are "confined" in catchments anyway. Isn't this the very idea of catchments to have all waters included in one larger scale area?

Agreed, catchment scale analysis is a good and proven approach, and fluxes are "confined" to catchments. However, variability in catchment response to perturbation, which is potentially indicative of variability in hydrochemical process, is difficult to evaluate in a robust manner without an approach which assesses multiple catchments/regions in a harmonized way. Text will be clarified.

L49     Yes, catchment waters will be influenced by land cover and geology, but so are observation on larger scales.

Agreed; text will be clarified to refer to regions (i.e., areas with differing land use/geology) as opposed to catchments: "For example, with freshwater acidification, water chemistry response to acid deposition may be altered by geology and land use/land cover, thus observations made in one watershed/region

may not generalize to others." Past research conducted in eastern North America has shown different drivers of freshwater acidification response, even in watersheds with similar land cover and geology (e.g., Hayes and Anthony, 1958; Rotteveel and Sterling, 2020).

L51     "affected"

Acknowledged; text will be updated.

L79     I understand the point that the authors want to make here, however, the example may be a bit too tightly defined. Looking for "water chemistry database sweden" yields the website of the water information system VISS (https://viss.lansstyrelsen.se), I don't know if data is extractable there but it seems that it is a good starting point. With this approach and a slight variation in search terms, more data should be discoverable.

Agreed; example removed.

L94     Can you state how much data was discarded, in %? Maybe leave the data in the dataset but provide a flag so that users can decide based on their needs?

Agreed, low quality data points will be flagged, not removed. Information on the proportion of low quality data points will be added.

L107     "simplied"

Acknowledged; text will be updated.

L107     2x "reduce storage requirements"

Acknowledged; text will be updated.

L129     Replace "standardized" with "harmonized" as probably most coordinates adhere to some kind of standard.

Agreed; text will be updated.

L146     Cost may be one reason but also, these are the most relevant parameters for many fields of research.

Agreed; text will be updated.

L148     What do you mean with under-reported results? Unclear.

Agreed, text will be clarified to state "What is currently unknown, is if analysis results are under-reported for some variables; that is, if all laboratory analysis results are reported for each sample included in the input databases".

L150     If no location data, I can understand the point. But w/o method information, it could still be interesting data in a global context (see argument above).

Acknowledged; these data cannot be included in SWatCh because the data do not adhere to the DS-WQX data schema due to missing variable fractionation information.

L156     Unclear logical connection between data gaps and discharge dependency.

Agreed, text will be clarified to state "The lower data coverage in these regions may inhibit the detection – and therefore treatment – of emerging climate change induced water quality problems, as the concentrations of many water chemistry variables are discharge dependant (Moatar et al., 2017)."

L168     "people"

Acknowledged; text will be updated.

L168     "who collected" -> "collecting"

Acknowledged; text will be updated.

---

## Author Response (AR1)

**Response to Referee Comments**

The Surface Water Chemistry (SWatCh) database: A harmonized global database of water chemistry to facilitate large-sample hydrological research

| Comment Number | Referee Comment | Response to Referee Comment | Change to Manuscript |
|---|---|---|---|
| Community Referee Mary Kruk | | | |
| 1 | Hello, With your manuscript currently under review I thought this would be a good opportunity to highlight a Canadian water quality database that currently addresses some of the challenges outlined in your paper. I work on DataStream, an open access platform for sharing water quality data. It allows users to access, visualize, and download water quality datasets collected by monitoring programs across regional hubs in Canada (the Mackenzie Basin, Lake Winnipeg Basin, Atlantic Canada, and a Great Lakes hub coming October). A main focus of DataStream is to help community monitoring groups, citizen scientists, researchers, and governments share their data at a regional-scale by adopting the US EPA/USGS WQX data standard to promote data (re)use and interoperability in transboundary watersheds. We thought it would be relevant to reach out because DataStream has faced many of the same challenges you address in your paper -- such as differing sample collection/analytical methods, reporting ambiguity, and spatial data gaps across Canada. We have found that the adoption of the WQX schema, used in the US Water Quality Portal, has helped us to align data collected by a wide range of monitoring initiatives. DataStream requires metadata on sample collection and analytical methods with each data point and reduces variable naming ambiguity by | Thank you very much for taking the time to review our manuscript and provide feedback. We enjoyed the off-line discussions regarding the points you raised. After reviewing the DS-WQX and US EPA/USGS WQX schema, we agree that the DS-WQX schema is highly suitable for SWatCh and will increase the inter-usablity of SWatCh with other existing large-sample datasets. Adapting SWatCh to conform to the DS-WQX schema also addresses several of Reviewer 1 and 2 comments, as discussed below. Based on these justifications, we will adapt SWatCh to conform to the DS-WQX schema. | SWatCh re-formatted and validated against DS-WQX schema. |

| Comment Number | Referee Comment | Response to Referee Comment | Change to Manuscript |
|---|---|---|---|
| | using the WQX list of allowed values for water chemistry parameters. We are constantly trying to evolve and improve the DataStream data standard and platform to better address these issues as I'm sure you are aware it is a large undertaking. Given the alignment between your area of research and our work with DataStream I would encourage you to review the DataStream schema (https://github.com/gordonfn/schema) for consideration in your manuscript.
Sincerely,
Mary Kruk | | |
| Anonymous Reviewer #1 | | | |
| 2 | The manuscript by Rotteveel and Sterling presents the global surface water chemistry (SWatCh) database, which contains data for 17 variables (Al, Fe, major ions, nutients, organic C, pH, etc.) from 9 million samples collected between 1960 and 2019. This database has the specific purpose to support research on surface water acidification. To create this database, the authors used data from 6 exiting hydrochemical databases/dataset, which they put in a uniform format, and then removed samples that were flagged as problematic and duplicates that exist as some of the databases used have culled data from the other databases.
I was able to download and use SWatCh without any problem. The download process is straight forward, and the database is easy to use.
While it is a very important task to assemble available data into such a large, publically available database, I feel that the authors have done a very poor job with regard to quality checks. They only discarded data that was already flagged as problematic, or which had very clearly unrealistic values, which was limited to negative values for concentrations. I think a much more robust quality check would be required to publish this dataset with an article in ESSD. Further, I feel that the authors did a rather poor job at analysing and presenting the data. For these two points, please see my major comments | Thank you very much for your thorough review of our manuscript and your feedback regarding the content, quality, and presentation of the database.
We have addressed your comments on database quality, analysis, and presentation below. | Changes to the manuscript are made in response to the specific comments below. |

| Comment Number | Referee Comment | Response to Referee Comment | Change to Manuscript |
|---|---|---|---|
| | further below. | | |
| 3 | Finally, I would like to highlight that this database does not contain any data on alkalinity, acid neutralisation capacity (ANC), DIC or HCO3- concentrations. This information can be found in at least a few of the databases from which data was taken for SWatCh. More importantly, these parameters represent the buffering capacity of a surface water body against acidification, and would thus be of huge importance for the study of surface water acidification and recovery. It is completely incomprehensible for me why these parameters were not included in SwatCh. | We recognize the importance of alkalinity, acid neutralization capacity (ANC), dissolved inorganic carbon (DIC), and bicarbonate ($HCO_3^-$) in determining surface water buffering capacity. Alkalinity and ANC are calculated parameters, and reported results may differ based on the calculation, for example, alkalinity may be expressed as total, hydroxide, carbonate, or bicarbonate alkalinity but reported in the input dataset as "alkalinity". Since sample fractionation and analysis/ calculation methods were often omitted in the input datasets, we originally elected to omit these parameters due to the uncertainty associated with the reported values. | Alkalinity, ANC, DIC, $HCO_3^-$, carbonate ($CO_3^{2-}$), and $CO_2$ added to SWatCh. |
| 4 | I suggest that major revision are necessary before this study can be considered for publication in ESSD. Please, see my major and general comments below: | We have addressed your comments on database quality, analysis, and presentation below. | Changes to the manuscript are made in response to the specific comments below. |
| 5 | Major comment #1: Quality checks of database
You should check all parameter values if they are reasonable, even if they are not flagged. You should check for instance for unreasonable high values, which can be due to mistakes made with the units (in particular for a database like GloRiCh, where data was assembled from lots of different dataset). If for instance mg and ug (or mM and uM) have been mixed up at some point (that could already be a mistake in the dataset you are taking data from), this might lead to errors of three orders of magnitude. I would suggest to first define for each parameter a realistic value range. Then, for all values lying outside of that range, you should first check is that concerns only one value in a time series, or the whole time series of a sampling site, or all values of a certain data source (note that for instance GloRiCh | Agreed. In addition impossible data (i.e., negative values for all parameters other than temperature, alkalinity, or ANC) being flagged as "Rejected", data are validated in the following additional ways:

1. Data are validated using the Data Stream (DS)-water quality exchange (WQX) data schema, a simplified adaptation of the United States Environmental Protection Agency (US EPA) WQX schema. The WQX schema is an implementation of the Environmental Sampling, Analysis, and Results (ESAR) data standard (EPA, 2006) which conforms to the | SWatCh is updated to conform to DS-WQX schema. Description of data standardization to, and validation against, the DS-WQX schema Additional data validation step 1 added to Section 2.3.1 (Database |

| Comment Number | Referee Comment | Response to Referee Comment | Change to Manuscript |
|---|---|---|---|
| | gives references of the data sources it used, and already in GloRich such mistakes might be present). If extreme values concern one specific sampling location, it might be worth investigating if that might be due to an exceptional site. For instance extremely high F- concentrations might be due to hydrothermal influence. Extreme PO4- concentrations can be due to phosphate deposits, like in the Peace River catchment, Florida. Sediments from dried out lakes might yield high concentrations in NaSO4. Etc. For each sampling location you should look at the time-series and try to identify potential outliers within the time series. For each outlier, you might want to check if other parameters are also affected, which could mean that either something exceptional has happened or that data from another sampling location has been wrongly attributed. Anyway, you should flag those values. You cannot assume that all suspicious data has already been flagged accordingly, in particular as data comes from very different sources, and some of them, like GloRiCh, are again assembled from different sources with different degree of quality checks. Concidering the size of the database and the number of included sites (38,598), this can realistically only be achieved programically. Thus, we would have to pre-define a statistically valid method of identifying outliers and remove/flag them based on those results. | National Environmental Information Exchange Network (EN) standards (EPA, 2022). The schema are standardized data formats which specify the data elements and dataset structures. Additional details on the schema are provided in the manuscript. The schema do not include screening for realistic values; we have addressed this gap by including the following two additional validation steps.

2. Each timeseries in SWatCh is screened for potential outliers using a four-times median absolute deviation (MAD) cut-off value. MAD is preferred to other methods of outlier removal when the data have a skewed distribution and is more robust against large outliers, which are present in SWatCh (Leys et al., 2013; Rousseeuw & Hubert, 2011). Outliers are flagged in the "ResultComment" column. | Format).

Section 2.2.3 (Flagging of Potential Outliers) added to describe additional data validation Step 2.

Values greater than four times the MAD are flagged as "potential outlier, value greater than four times the median average deviation" in the "ResultComment" column.

Values less than four times the MAD are flagged as "potential outlier, value less than four times the median average deviation" in the "ResultComment" column. |
| 6 | Major comment #2: Presentation of database
Your results section is very short, and your discussion section doesn't make many links to your own results. Figure 2 is a good beginning to represent the available data, but it would be more interesting if the spatial coverage was represented separately for | Agreed, Figure 2 has been expanded to demonstrate the spatial data coverage by water body type. | Figure 2 updated.

Results section expanded as per below. |

| Comment Number | Referee Comment | Response to Referee Comment | Change to Manuscript |
|---|---|---|---|
| | different types of inland water bodies. It is not clear at all from your manuscript how well lakes vs. reservoirs vs. rivers are represented in tat database. | | Discussion section updated to make more links to the results section. |
| 7 | It would also be interesting to know the numbers of samples per water body type that have measurements for a specific combination of parameters that are interesting with regard to acidification, like: How many samples are there with all major ions and pH? Here you should maybe start with an overview of which combinations of parameters are usually used to study acidification. I guess samples where only sodium or phosphate was measure are not that interesting. Maybe you can make a ranking of parameter combinations that allow you to study acidification with a different degree of conclusiveness and certitude. And then list the number of samples that have measurements for these parameter combinations, and do that separately for different kinds of water bodies (lakes, reservoirs, canals, ditches, rivers, etc.) and different world regions (at least continents, or major biomes/climate zones). You should also give an overview about which time-periods are covered in different parts of the world. That would very important if you want to investigate temporal trends in acidification recovery. You should also think about presenting data density (number of sites, number of samples per site, average length and frequency of time-series, etc. ) for different types of inland waters as a map. You could take inspiration from figure 2a in Regnier et al. 2013 (Nature Geoscience, DOI: 10.1038/ngeo1830), that created a density index for pCO2 values. | Agreed. We have updated/added the following tables and figures to the results and discussion sections.

1. Table 3 has been expanded to include information on the date ranges and sampling frequency of the timeseries for each parameter.

2. Table 4 has been added to the results section. Table 4 lists the number of sites for which a given set of variables (relevant to the study of freshwater acidification) are available by timeseries length and water body type.

3. Figure 3 has been added to the results section. Figure 3 presents the spatial distribution of the sites in each category in Table 4. | Results section expanded with update of Table 3, and addition of Table 4, Figure 3, and associated text.

Additional discussion added to Section 4.1 related to the results presented in Table 3, Table 4, and Figure 3. |
| 8 | You should also take into account global geodatasets that allow for regional classification of water bodies, like for instance the HydroAtlas (Linke et al. 2019, https://doi.org/10.1038/s41597-019-0300-6). Like this you could make more qualified statement about which kind of river or lake is underrepresented in your dataset. You state you cannot link your chemistry data | Disagreed; we do not state that linking SWatCh to catchment properties is not possible. We linked the sites to catchment properties to examine the distribution of the sites with respect to geology using the USGS World Geologic Maps datasets | Section 4.4 revised to include reference to HydroATLAS. |

| Comment Number | Referee Comment | Response to Referee Comment | Change to Manuscript |
|---|---|---|---|
| | to catchment properties, but with HydroAtlas you could get a good idea what kind of river-catchment systems are well represented and what kind underrepresented. | (https://certmapper.cr.usgs.gov/data/apps/world-maps/), land use type using the HYDE datasets (doi:10.1111/j.1466-8238.2010.00587.x), and climate zones using the Köppen-Geiger climate classification map (https://certmapper.cr.usgs.gov/data/apps/world-maps/), as stated in the results section. We are unable to include catchment information due to DS-WQX schema restrictions. Information provided in HydroATLAS would be very useful in the development of global water chemistry models using SWatCh; thus, we have referred the reader to this resource in the Limitations and Future Work section. | |
| 9 | General comments:
L74-76: How did you perform those checks? Where can I see the results? I think a quality assessment of this kind is very important. | Acknowledged. The spot-checks we refer to were a high-level review of the methodology used for sample analysis; for example, by verifying that an approved analysis method was used. Since sample certificates of analysis are not available, additional verification of laboratory quality assurance/quality control procedures is not possible. We have removed the statement from the manuscript.
The validation of the data has been improved, as described in response to comment 5. | Statement removed from manuscript. |
| | L85-86: I wonder how you identified these "untreated" water bodies. I know that in GloRiCh this information is not given. Here, some analysis of the water chemistry data itself could have been useful to spot suspicious cases, for which some investigation could have been performed based on the location information. | Acknowledged. As stated in the manuscript, we define "untreated" as "water that is not wastewater or receiving treatment plant effluent near to the sample collection site." For example, we exclude sample locations identified as "wastewater" or "effluent"; this information is provided in the input datasets. | Statement clarified and expanded. |
| 10 | L88: By "phosphorus", do you mean "total phosphorus"? | Acknowledged. We include both total and dissolved phosphorus; hence, no fractionation is identified here; included sample fractions are | Statement on included sample fractions added to |

| Comment Number | Referee Comment | Response to Referee Comment | Change to Manuscript |
|---|---|---|---|
| | | presented in Table 3. | Section 2.2 (Data Inclusion). |
| 11 | L91-92: Error message instead of reference. | Acknowledged. Removed. | Error removed. |
| 12 | Section 4.2: When discussing these effect of methodological changes on time-series data, you should combine that indeed with an analysis of at least the longest time-series you have in your database. | Agreed, an example demonstrating the effects of methodological changed on timeseries interpretation has been added (Figure 4 and associated discussion.) | Figure 4 and associated discussion added to Section 4.2 |
| 13 | Section 4.4: Here you mention that you often do not have the discharge data associated to the water chemistry data. Did you try to match the river water sampling locations with stream gauges from the Global Runoff Data Centre (GRDC, https://www.bafg.de/GRDC/EN/Home/homepage_node.html)? | Acknowledged. The Global Discharge Data Centre includes approximately 10,000 stations, which is approximately half of the 21,089 river sampling locations in SWatCh (assuming each stream/river location in SWatCh is included in the Global Discharge Data Centre dataset. As stated in the manuscript, integrating discharge data into SWatCh is considered an area for future development, and beyond the current scope of work. | Reference to Global Runoff Data Centre Added. |
| Anonymous Reviewer #2 | | | |
| 14 | The authors present a newly created database on chemical composition of surface waters. The database is comprised of several database sources from which specific parameters/variables are extracted and unified for the specific purpose to provide a data base for surface water acidification research. The collection and harmonization of data on water chemistry is very important to the research community, as it enables more refined global analyses of matter fluxes, temporal developments, climate change impacts, any many more. The manuscript addresses an important data topic, which makes it worth to be published. However, due to the points stated below, I recommend a major revision. | Thank you for your detailed review of our manuscript. We appreciate your feedback regarding data quality, harmonization, and selection, database structure and presentation, and text quality. We have responded to your comments below. | No change required. |
| 15 | Data quality I would argue, from a personal viewpoint, that if the goal is to provide global coverage of data to enable global cross- | Agreed, we have endeavoured to strike the balance between your comments and those of Reviewer 1 by flagging values greater or less | Section 2.2.3 (Flagging of Potential Outliers) |

| Comment Number | Referee Comment | Response to Referee Comment | Change to Manuscript |
|---|---|---|---|
| | boundary evaluation of surface waters, it may not be very important to have a high data quality, as the available amount of data will level out "outliers" or differences in the data analyses from a statistical viewpoint. | than four times the MAD, but not removing those values. This approach allows users to handle potential outliers in a manner which is suitable for their specific purpose. Please see associated response to comment 5. | added. Please see associated changes made under comment 5. |
| 16 | Data harmonisation
The calls for a unified approach in all future data collections are very noble, but I doubt that they will be heard. Data producing authorities very often have their own, historically grown structures and formats, that are so convoluted and unpredictable that it would be and hopeless to expect a globally unified data structure | Acknowledged. Although this comment does not request a change to SWatCh, we have updated the format of SWatCh to adhere to the DS-WQX schema, as suggested by the Community Reviewer. By making this change, we aim to contribute to a more harmonized approach. | No change required. |
| 17 | Data selection
The authors state that the parameters were specifically selected to evaluate surface water acidification, however I would argue that the most important parameter in this regard is missing: total alkalinity (TA). This is reported in some of the used sources, even if it may be in awkward units sometimes. The TA is fundamental for the understanding of the carbonate system and the interaction of CO2 and natural waters. Alternatively, dissolved inorganic carbon could be included, or both parameters, where available, to be able to calculate the missing parts of the carbonate system (TA and pH or DIC and pH enable the calculation of DIC or TA, respectively). Furthermore, the inclusion of TA would enable the calculation of a charge balance, which could provide an indicator for the data quality. | Agreed, please see our response to Comment 3 above, which addresses the same issue. | Alkalinity, ANC, DIC, $HCO_3^-$, carbonate ($CO_3^{2-}$), and $CO_2$ added to SWatCh. |
| 18 | Database structure and presentation
I really appreciate the approach of publishing the scripts for the database of Github. This makes the work very transparent and should be an example for all scientists working with complex data processing. | Acknowledged; thank you. | No change required. |
| 19 | The chosen format of the data is slim and straightforward, however, for the average enduser, the relational style of the files | Acknowledged. The data format has been simplified to adhere to the DS-WQX format | No changes made. |

| Comment Number | Referee Comment | Response to Referee Comment | Change to Manuscript |
|---|---|---|---|
| | may present a potential problem as data cannot be filtered and used as is, but have to be transformed. It may be an advantage (not a requirement) to provide a python script that converts the data into the "classical" column-row-format. It may, however, increase the filesize to an extent that makes it hard to handle. | (i.e., one file, as opposed to three relational databases). Based on the file size of SWatCh (i.e., 1.8 GB), it is already too large to read into a spreadsheet program such as Excel without truncating the dataset. The file size of the combined and expanded (i.e., column-row format) format would likely be an order of magnitude larger, would cause similar issues, and remain unusable for an end user unfamiliar with programmatically processing data. Based on this. We have not provided an additional data processing script. | |
| 20 | Regarding the units, the choice of weight units is okay but may lead to the need to recalculate to molar units as this is needed in geochemical calculations (e.g., charge balance, ratios, chemical formulas). | Acknowledged. | No changes required. |
| 21 | Text quality
There are several typos, duplications and wording issues in the text. I mention some of them below. Overall, the text could benefit from a revision, which clears out the errors but more specifically narrows focus on the specific arguments for the need of a new and harmonized database. | Agreed. The text will be reviewed for errors prior to re-submission. By addressing your specific comments below regarding L47 below, we aim to clarify why a harmonized water chemistry database is required to answer global-scale surface water acidification research questions. | Text revised for grammatical clarity.

Additional justification added regarding the need for a database allowing transboundary analysis. Please see associated change under Comment 29 below. |
| 22 | Specific comments
L8      2x "identify" | Acknowledged. | Revised. |
| 23 | L18      Define the need for more data collection – how would | Agreed; we do expect to see a lower number of | Sentence re- |

| Comment Number | Referee Comment | Response to Referee Comment | Change to Manuscript |
|---|---|---|---|
| | that improve global models? Little data from arid regions may also be due to the fact that there are less surface waters | samples in arid regions. Prior research has indicated that additional data are required for some the regions we have identified as having lower data availability to improve global models of water chemistry (e.g., Harrison et al., 2005; Harrison, Caraco, and Seitzinger, 2005). | phrased to refer to the African and Asian continents. Associated change in Section 4.1. |
| 24 | L19    "Environs" | Unclear comment. "Environs" is defined as "environing things: surroundings" or "an adjoining region or space: vicinity", and is thus suitable for the context in which it is used. | No changes made. |
| 25 | L21/22  2x "address | Acknowledged. | Revised. |
| 26 | L29    "a number projected…" is meant to refer to the 4 bln people, but as it stands in the text it rather refers to "at least one month" | Acknowledged. | Text revised for clarity. |
| 27 | L30    "these resources" – which? | Acknowledged. | Text revised for clarity. |
| 28 | L36    Define "transboundary problem" | Acknowledged, in the context of this manuscript we define a "transboundary problem" to be a water quality issue, or cause of a water quality issue, which crosses international borders. For example, a main driver of freshwater acidification in Atlantic Canada is acid deposition originating from all the major production regions in North America, including those in the United States of America (Shaw, 1979). A similar definition of "transboundary problem" is often used when discussing water availability issues which cross international borders (e.g., Thu and Wehn, 2015). | Definition added to text. |
| 29 | L47    When I comes to the fate and behavior of compounds in natural water, I would argue, the catchment scale is a good and proven approach. I may not understand the term "transboundary" in your sense, but why should be look | Agreed, catchment scale analysis is a good and proven approach, and fluxes are "confined" to catchments. However, variability in catchment response to perturbation, which is potentially | Text clarified. |

| Comment Number | Referee Comment | Response to Referee Comment | Change to Manuscript |
|---|---|---|---|
| | transboundary if fluxes are "confined" in catchments anyway. Isn't this the very idea of catchments to have all waters included in one larger scale area? | indicative of variability in hydrochemical process, is difficult to evaluate in a robust manner without an approach which assesses multiple catchments/regions in a harmonized way. | |
| 30 | L49    Yes, catchment waters will be influenced by land cover and geology, but so are observation on larger scales. | Agreed; text will be clarified to refer to regions (i.e., areas with differing land use/geology) as opposed to catchments: "For example, with freshwater acidification, water chemistry response to acid deposition may be altered by geology and land use/land cover, thus observations made in one watershed/region may not generalize to others." Past research conducted in eastern North America has shown different drivers of freshwater acidification response, even in watersheds with similar land cover and geology (e.g., Hayes and Anthony, 1958; Rotteveel and Sterling, 2020). | Text revised. |
| 31 | L51    "affected" | Acknowledged. | Revised. |
| 32 | L79    I understand the point that the authors want to make here, however, the example may be a bit too tightly defined. Looking for "water chemistry database sweden" yields the website of the water information system VISS (https://viss.lansstyrelsen.se), I don't know if data is extractable there but it seems that it is a good starting point. With this approach and a slight variation in search terms, more data should be discoverable. | Agreed. When the data was originally collected, the search terms did not return a result. It appears that the search terms of the website may have been updated. | Example removed. |
| 33 | L94    Can you state how much data was discarded, in %? Maybe leave the data in the dataset but provide a flag so that users can decide based on their needs? | Agreed. Data processing updated to flag, not remove, low quality data. | Low quality data points flagged.

Information on the proportion of low-quality data points added. |

| Comment Number | Referee Comment | Response to Referee Comment | Change to Manuscript |
|---|---|---|---|
| 34 | L107 "simplied" | Acknowledged. | Revised. |
| 35 | L107 2x "reduce storage requirements" | This section of the text has undergone major revisions and this sentence has been removed. | No changes made. |
| 36 | L129 Replace "standardized" with "harmonized" as probably most coordinates adhere to some kind of standard. | Agreed. | Revised. |
| 37 | L146 Cost may be one reason but also, these are the most relevant parameters for many fields of research. | Agreed. | Revised. |
| 38 | L148 What do you mean with under-reported results? Unclear. | Acknowledged. | Clarified. |
| 39 | L150 If no location data, I can understand the point. But w/o method information, it could still be interesting data in a global context (see argument above). | Acknowledged; these data cannot be included in SWatCh because the data do not adhere to the DS-WQX data schema due to missing variable fractionation information. | Revised. |
| 40 | L156 Unclear logical connection between data gaps and discharge dependency. | Agreed. | Clarified. |
| 41 | L168 "people" | Acknowledged. | Revised. |
| 42 | L168 "who collected" -> "collecting" | Acknowledged. | Revised. |

---

## Author Response (AR2)

**Response to Comments**

The Surface Water Chemistry (SWatCh) database: A harmonized global database of water chemistry to facilitate large-sample hydrological research

| Comment Number | Referee Comment | Response to Referee Comment | Change to Manuscript |
|---|---|---|---|
| 1 | Referee II (report) suggested to merge results and discussion together to extend the chapter 3 on results, we suggest that this is not needed as two Figures (Figures 2 and 3) and 2 tables (Tables 2 and 3) will be in the chapter 'results'. | We have ensured that the results and discussion are not merged together. | No change |
| 2 | minor edits (line numbers refer to the tracked change document)
Abstract L15 'by six national and international agencies -> please list the agencies/data sources in the abstract, please add Canadian to 'national agencies' throughout the text. | The agencies/data sources have been added to the abstract. | Revised as suggested |
| 3 | The short names of the data sources / agencies that are shown in Figure 2 could be introduced in listing the long and short names in chapter 2, L89/90, or you could add all short names in Table 1. | The short names have been added to Table 1. | Abbreviated names added to Table 1 |
| 4 | Reference list – please add all DOI referenced data sets (e.g. table 1, data sources) to the reference list | The DOI-referenced datasets in Table 1 have been added to the reference list. | Revised as suggested |